# Video Prediction via Selective Sampling

**Jingwei Xu,    Bingbing Ni,**\*    **Xiaokang Yang**
MoE Key Lab of Artificial Intelligence, AI Institute
SJTU-UCLA Joint Research Center on Machine Perception and Inference,
Shanghai Jiao Tong University, Shanghai 200240, China
Shanghai Institute for Advanced Communication and Data Science
xjwxjw,nibingbing,xkyang@sjtu.edu.cn

## Abstract

Most adversarial learning based video prediction methods suffer from image blur, since the commonly used adversarial and regression loss pair work rather in a competitive way than collaboration, yielding compromised blur effect. In the meantime, as often relying on a single-pass architecture, the predictor is inadequate to explicitly capture the forthcoming uncertainty. Our work involves two key insights: (1) Video prediction can be approached as a stochastic process: we sample a collection of proposals conforming to possible frame distribution at following time stamp, and one can select the final prediction from it. (2) De-coupling combined loss functions into dedicatedly designed sub-networks encourages them to work in a collaborative way. Combining above two insights we propose a two-stage framework called VPSS (**V**ideo **P**rediction via **S**elective **S**ampling). Specifically a *Sampling* module produces a collection of high quality proposals, facilitated by a multiple choice adversarial learning scheme, yielding diverse frame proposal set. Subsequently a *Selection* module selects high possibility candidates from proposals and combines them to produce final prediction. Extensive experiments on diverse challenging datasets demonstrate the effectiveness of proposed video prediction approach, *i.e.*, yielding more diverse proposals and accurate prediction results.

## 1   Introduction

Video prediction has been receiving increasing research attention in computer vision [3, 29, 10, 8, 19], which has great potentials in applications such as future decision, robot manipulation and autonomous driving [20]. Previous methods [10, 19] with pixel-wise regression loss tend to produce blurry results as they seek the average from possible futures [29]. To enhance the generation quality, some works [8, 34] utilize adversarial learning [13] to facilitate video prediction task, *i.e.*, adding an adversarial loss [13] on the prediction module.

However, paired regression and adversarial loss still CANNOT solve image blur and motion deformation problems in principle. A generator often struggles to balance between adversarial [13] and regression loss during training procedure [18, 2, 5], thus most possibly yielding an averaged result. As in worse case either adversarial [13] or regression loss tend to take dominate place, which forces the other term to fail to play its role. In other words, both loss functions work rather in a competitive way than collaboration. We start to think: (1) To address the blur issue, is it possible to sample a collection of high quality proposals conforming to possible frame distribution at following time stamp, and select the final prediction from it? (2) To encourage collaboration between loss functions, is it possible to design dedicated sub-networks for adversarial and regression loss respectively?

---

Table 1: Competition between adversarial and regression Loss in term of $\lambda \mathcal{L}_{Adv} + \mathcal{L}_{Reg}$.

| Model | MCNet [34] $\lambda$=0.02/0.2/0.5 | DrNet [8] $\lambda$=1e-4/1e-3/4e-3 | SAVP [25] $\lambda$=0.01/0.1/0.3 |
|---|---|---|---|
| $\mathcal{L}_{Reg}$ after 50K iterations | 0.04/0.05/0.07 | 0.07/0.09/0.12 | 0.02/0.03/0.05 |

Motivated by these issues, we propose a two-stage framework called VPSS which utilizes different modules to handle both losses respectively, therefore to encourage collaboration between them, instead of competition. As shown in Figure 1, the *Sampling* module produces multiple high quality video frame proposals, by making use of a multiple choice adversarial learning scheme, yielding diverse video prediction set. This module is trained in an adversarial learning manner [5]. The *Selection* module selects high possibility candidates from proposals and combines to produce the final prediction, according to the criteria of better position matching. By contrast, the selection module is trained with regression loss. We conduct both qualitative and quantitative experiments on diverse datasets, ranging from digits moving, human motion to robotic arm manipulation, including a visualization experiment. Our results clearly indicate higher visual quality and more precise motion prediction, even for complex motion patterns as well as long-term prediction, which significantly outperform prior arts. Experiments show that these two modules could corporate well with each other under the VPSS framework.

## 2   Related work

**Video Prediction**. Many previous work have been done on video prediction task. Srivastava *et al*. [33] first proposes to use LSTM [16] to predict future frames. Shi *et al*. [32] proposes ConvLSTM architecture dedicated to preserve the spatial information, which is a big step forward on this task. Different from ConvLSTM, DFN [19] generates convolution kernels according to the inputs, which shows more flexibility on modelling the motion variation. CDNA [10] combines the advantages of ConvLSTM [32] and DFN [19] for further utilizing the spatial information. However all above methods suffer from the image blur and structure deformation because of the regression loss. To enhance the image quality, DrNet [8] and MCNet [34] utilize adversarial training to generate more sharpen frames, yet hard to balance adversarial loss and regression loss. Some work [35, 20, 31] further push prediction model to achieve long-term prediction. More recently, stochastic prediction [7, 3, 25] tries to use random learned prior or noise to achieve prediction in a stochastic way, which generate visually appealing prediction but with stochastic motion prediction (*i.e.*, location variation and shape transformation is random). Note that different from these stochastic prediction methods, our model first generates multiple proposals and then precisely infer the final outcome from them, which shows high precision motion prediction.

**Competition between Adversarial and Regression Loss**. This issue has been widely discussed in image generation and translation task [18, 36]. It is more commonly treated as the competition between generation diversity (*i.e.*, related to GAN [13] loss) and visual quality (*i.e.*, related to regression loss) in image domain. VAE-GAN [24] first proposes to combine VAE and GAN [13] by replacing element-wise errors with feature-wise errors to better capture the data distribution. Some work try to solve this problem from information entropy view, such as InfoGAN [6], ALI [9] and ALICE [26], which mainly try to find a deterministic mapping relation between the image domain and latent code domain. Based on VAE-GAN [24], BEGAN [4] proposes a new equilibrium enforcing method to pursue fast and stable training and high visual quality. Different from the image generation task, video prediction further requires precise motion prediction along the time axis. Accordingly, the competition in video domain lies between visual quality and motion prediction precision, yet few methods have been proposed to address this issue.

## 3   Method

Previous models [34, 34, 25] try to balance adversarial [13] loss (denoted as $\mathcal{L}_{Adv}$) and regression loss (denoted as $\mathcal{L}_{Reg}$) with a hyper-parameter $\lambda$, *i.e.* $\lambda \mathcal{L}_{Adv} + \mathcal{L}_{Reg}$. To demonstrate the competition intuitively, we run three open-source codes with their default training setups[2], only altering $\lambda$ with

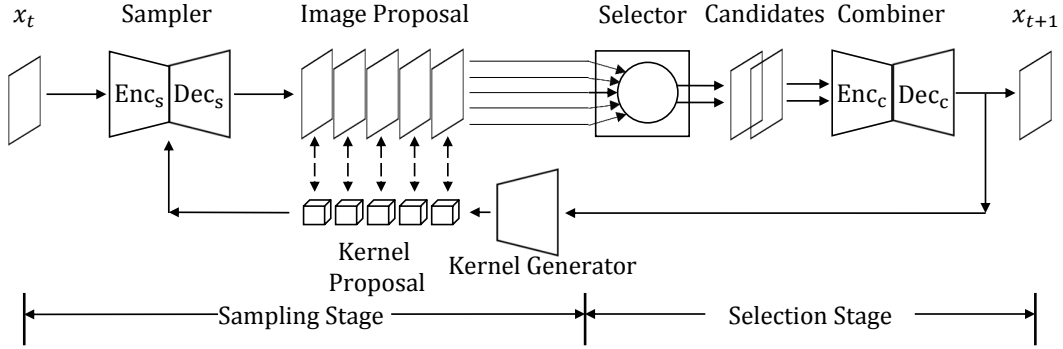

Figure 1: The general framework of VPSS. At the **Sampling** stage, our framework contains a conditional sampling module which produces $N$ high-quality proposals at time stamp $t + 1$ based on inputs at time stamp $t$. For the **Selection** stage, we propose a selection module for final prediction. This module further chooses $K$ candidates from the high-quality proposals and fuses them to produce final prediction.

different values. As shown in Table 1, if enlarging $\lambda$ progressively, one can notice that converge value of $\mathcal{L}_{Reg}$ keeps increasing. This indicates that it is impractical to force prediction module to satisfy both sides, which directly motivates us to explore this task from a different view. Specifically, we propose a novel framework called VPSS with two modules of sampling and selection. In this section we discuss this framework and detailed architecture of both modules. Meanwhile we demonstrate how these two modules effectively corporate well with each other.

### 3.1 Sampling Module

Previous methods [25, 3] aim to model future uncertainty with temporal correlated noise, which is sampled from prior distribution then passed through a recurrent neural network (*e.g.*, GRU, LSTM). However it is insufficient to model the upcoming uncertainty explicitly with random noise based stochasticity, which lacks correlation with inputs. Inspired by previous work [30, 5], we show that video prediction can be approached as a stochastic process: we gather a random collection of high quality proposals in one shot, with a multiple choice adversarial learning scheme that encourages diversity within the collection.

More formally, consider single step forward prediction, where we use current input (denoted as $\mathbf{x}_t \in \mathcal{R}^{L \times H \times C}$) to predict next time-stamp frame (denoted as $\mathbf{x}_{t+1} \in \mathcal{R}^{L \times H \times C}$). $L, H, C$ stand for image width, height and channel respectively. Recall that $\mathbf{x}_t$ is firstly fed into a sampler $\phi_{spl}$ to produce a collection of proposals (denoted as $\hat{\mathbf{X}}_{t+1} = \{\hat{\mathbf{x}}_{t+1}^1, ..., \hat{\mathbf{x}}_{t+1}^N\}$), where $N$ is the desired number of proposals. For single prediction the number of final output channels in $\phi_{spl}$ is $C$. So for $N$ proposals we directly enlarge it to $C \times N$, where each consecutive $C$-tuple of channels forms a proposal, and denote the corresponding kernel as $W = \{W^i\}_{i=1}^N$.

To guarantee the visual quality of proposals, we utilize adversarial learning to facilitate the sampling procedure:

$$\mathcal{L}_{spl}(S, D) = \sum_{t=1}^{T-1} (\mathbb{E}_{\mathbf{X}}[\log D(\mathbf{x}_{t+1})] + \frac{1}{N}\mathbb{E}_{\mathbf{X}}[\log D(1 - \phi_{spl}(\mathbf{x}_t, W))]), \qquad (1)$$

where $D$ is a discriminator capable of distinguishing sampled images from real images. Naturally we hope these proposals are diverse enough for further selection. However only using the same adversarial loss forces all proposals to be identical, which departs from the requirement of diversity. To this end, we propose a kernel generator $\phi_{KGen}$ to achieve this goal.

At time stamp $t$, the kernel generator takes previous $\mathbf{x}_t$, and $\mathbf{x}_{t-1}$ as input, and the output of $\phi_{KGen}$ is $\Delta W = \{\Delta W^i\}_{i=1}^N$. As shown in Figure 1 (black dash lines), there is one-to-one correspondence between $\Delta W^i$ and $\hat{\mathbf{x}}_{t+1}^i$. We denote the $m$-th input channel and $n$-th output channel of $\Delta W$ as

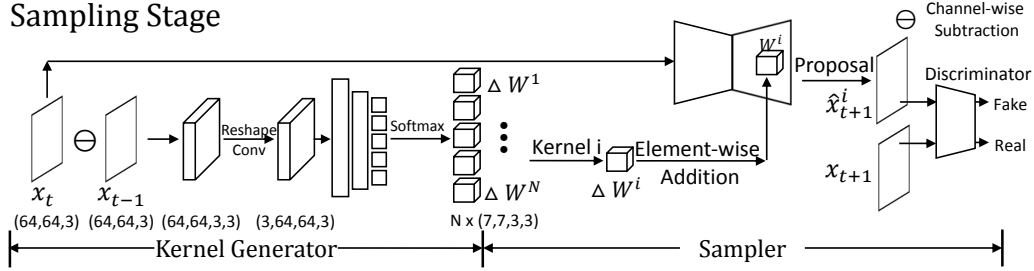

Figure 2: Detailed architecture of Sampling Module. The left part is proposed kernel generator to produce multiple kernels, while the right part is sampler for sampling a collection of high quality proposals.

$\Delta W(m,n)$. Similarly $\mathbf{x}(n)$ represents the $n$-th channel of $\mathbf{x}$ and $\Delta W(\cdot, n)$ represents the $n$-th output channel along with all input channels of $\Delta W$. As shown in Figure 2, the kernel generation procedure first performs channel-wise subtraction to obtain temporal variation information, and then encodes it into the current convolution kernel with a CNN $\phi_W$, Meanwhile we perform channel-wise softmax [28] along the input channel:

$$\Delta \hat{W}(m,n) = \phi_W(\mathbf{x}_t(m), \mathbf{x}_{t-1}(n)), \quad m, n = 1, ..., C, \tag{2}$$

$$\Delta W(\cdot, n) = Softmax(\Delta \hat{W}(\cdot, n)), \quad n = 1, ..., C, \tag{3}$$

where $C$ is the number of channels. The sampling procedure at time stamp $t$ is executed as follows,

$$\Delta W = \phi_{KGen}(\mathbf{x}_t, \mathbf{x}_{t-1}), \hat{\mathbf{x}}_{t+1}^i = \phi_{spl}(\mathbf{x}_t; W^i + \Delta W^i), \quad i = 1, ..., N. \tag{4}$$

Through kernel generator we actually transform this problem into kernel diversity, *i.e.*, diversity of $\{\Delta W^i\}_{i=1}^N$. Inspired from Guzman-Rivera *et al.* [14], we develop a multiple choice adversarial learning scheme as follows,

$$\mathcal{L}_{KGen} = \sum_{t=2}^{T-1} \mathbb{E}_{\hat{\mathbf{X}}_{t+1}}[\log D(\hat{\mathbf{x}}_{t+1}^k)], \quad k = \min_i ||\mathbf{x}_{t+1} - \hat{\mathbf{x}}_{t+1}^i||_1. \tag{5}$$

This loss function is a critical component in sampling module, and the diversity essentially results from the $\min$ operation in Equation 5. To be specific, at each time stamp we only update kernel corresponding to the best sampled image, where $N$ kernels ($\{\Delta W^i\}_{i=1}^N$) are optimized independently toward different directions throughout training procedure. This asynchronized updating scheme encourages the kernel generator to spread its bets and cover the exploration space of predictions that conform to all possible frame distribution. Intuitively one can consider that the kernel generator $\phi_{KGen}$ is boosted to capture motion information based on previous inputs and infer next time-stamp motion direction without constrain of regression loss. Meanwhile image quality is well guaranteed by adversarial learning loss (Equation 1). More importantly, different from those stochastic prediction methods [3, 7, 25] whose diversity is achieved by random noise, the diversity of our model is entirely achieved by previous inputs, which is more rational in principle and inherently useful for further selection.

## 3.2 Selection Module

Given sampled proposals $\hat{\mathbf{X}}_{t+1}$, the selection module first selects best K candidates from them under the measure of motion precision. For example, the motion direction usually changes smoothly between frames, which can be treated as a selection criterion. But it is impractical to hand-crafted design several criteria for candidate sifting. Meanwhile under the problem setting of video prediction, ground truth at time stamp $t + 1$ is unknown at time stamp $t$. Directly comparing with ground truth is not an option. To tackle this problem, we formulate it as ranking all proposals based on previous inputs. To be specific, for each proposal we utilize recurrent neural network to regress confidence score obtained in a self-supervised manner, and rank them based on the regressed values. The top-K candidates are feed into a combiner for further processing.

For clarity we denote the ground-truth score for $i$-th candidate as $\gamma_i$, and $\Gamma = \{\gamma_i\}_{i=1}^N$. Corresponding regressed score is denoted as $\hat{\gamma}_i$, and $\hat{\Gamma} = \{\hat{\gamma}_i\}_{i=1}^N$. The most direct way to generate ground-truth score

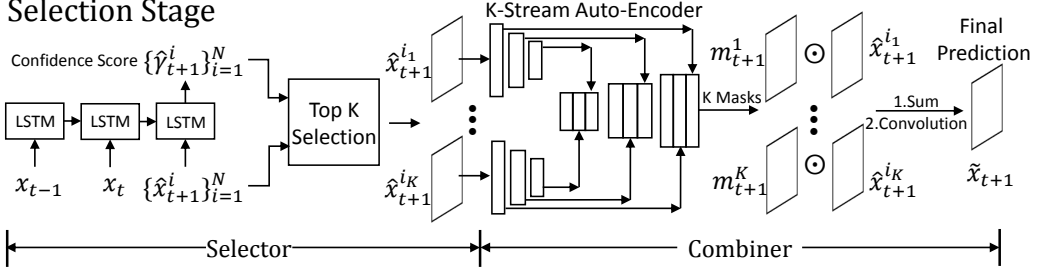

Figure 3: Detailed architecture of Selection Module. The left part is proposed selector to select $K$ candidates from $N$ proposals, while the right part is combiner for combining $K$ candidates into final prediction.

is to calculate the pixel-wise error between the proposals and ground truth: $\gamma_i = ||\mathbf{x}_{t+1} - \hat{\mathbf{x}}_{t+1}^i||_1$. It works well when the motion pattern is simple (*e.g.*, transformation toward some direction) and the objects are rigid bodies. When encountering real-world video sequences (*e.g.*, human or robot motion with occlusion), it severely suffers from the motion uncertainty and complex structure deformation.

For better capturing the motion and content information, we utilize a pre-trained discriminative model to extract the multi-level feature from video sequences, and compute the confidence score at feature level, which is widely used in image translation task [18]:

$$\gamma_i = \sum_{l=1} ||\varphi_l(\mathbf{x}_{t+1}) - \varphi_l(\hat{\mathbf{x}}_{t+1}^i)||_1, i = 1, ..., N, \tag{6}$$

where $\varphi$ is the discriminative model (*e.g.*, detection or pose estimation network), and $l$ stands for the $l$-th layer of $\varphi$. As mentioned above, $\Gamma$ is treated as target value of confidence score. Based on it we can predict the confidence score based on previous inputs as follows:

$$\hat{\gamma}_i = \psi_{slt}(\hat{\mathbf{x}}_{t+1}^i | \mathbf{x}_t, \mathbf{x}_{t-1}), \quad i = 1, ..., N, \tag{7}$$

where $\psi_{slt}$ is essentially a 3-step ConvLSTM network [32] (Left-most part in Figure 3). It successively takes $\mathbf{x}_{t-1}, \mathbf{x}_t, \hat{\mathbf{x}}_{t+1}^i$ as input . Therefore we propose to train the $\psi_{slt}$ with a regression loss: $\mathcal{L}_{slt} = ||\hat{\Gamma} - \Gamma||_1$. At time stamp $t$, we select top-K candidates $\tilde{\mathbf{X}} = \{\hat{\mathbf{x}}^{i_j}, i_j \in \mathcal{T}(\hat{\Gamma})\}_{j=1}^K$ for final prediction, where $\mathcal{T}(\hat{\Gamma})$ is the index set of top-K candidates in $\hat{\Gamma}$.

For the final step, we utilize a Combiner $\psi_{comb}$ to compose candidates $\tilde{\mathbf{X}}$ into final prediction. Correspondingly it requires $\psi_{comb}$ to fully capture information contained in $\tilde{\mathbf{X}}$. Inspired from recent work on video analogy making [31], we propose a K-stream Auto-Encoder to capture feature information. As shown in Figure 3, each candidate is fed into an encoder to extract multi-level information, then passed to decoder to generate K masks $\mathbf{M} = \{\mathbf{m}^j\}_{j=1}^K$, which are used to compose K candidates into final prediction:

$$\tilde{\mathbf{x}}_{t+1} = \psi_{comb}(\tilde{\mathbf{X}}) = Conv(\sum_{j=1}^K \mathbf{m}^j \odot \hat{\mathbf{x}}_{t+1}^{i_j}), \tag{8}$$

where $\odot$ represents hadamard product. As shown in Figure 3, we also use skip connection [18] between encoders and decoder to enhance feature sharing, and we keep the spatial resolution of feature same to images throughout $\psi_{comb}$, which proves more efficient to preserve the spatial information [21]. We use regular regression loss to train $\psi_{comb}$ as follows : $\mathcal{L}_{comb} = \sum_{t=2}^T ||\mathbf{x}_t - \tilde{\mathbf{x}}_t||_1$.

### 3.3 Design Considerations and Implementation Details

**Design Considerations**. Notably, our framework contains two modules and each involves two sub-networks. If without carefully designed training procedure, it will easily get stuck in a bad local minima with meaningless outputs. We therefore provide several practical considerations dedicated for training stability. Firstly, sampler $\phi_{spl}$ and selector $\psi_{slt}$ act as models with strong prior knowledge, pre-training them stabilizes following training procedure by a large margin. Secondly, we utilize curriculum learning [22] along the temporal direction, which could effectively flatten the training curve. Last but not least, we decouple combined loss function into corresponding sub-networks, *i.e.*, we update $\phi_{KGen}, \phi_{spl}, \psi_{slt}, \psi_{comb}$ according to $\mathcal{L}_{KGen}, \mathcal{L}_{spl}, \mathcal{L}_{slt}, \mathcal{L}_{comb}$ respectively. On one

Table 2: Detailed Evaluation Setup with different datasets and models.

| Datasets | Comparison models | Inputs and outputs | Video resolution |
|---|---|---|---|
| MovingMnist [33] | DFN [19]& SVG [7] | 10 inputs for 10 outputs | 64x64 |
| RobotPush [10] | CDNA [10]& SV2P [3] | 2 inputs for 8 outputs | 64x64 |
| Human3.6M [17] | DrNet [8]& MCNet [34] | 5 inputs for 10 outputs | 64x64 |

Table 3: Qualitative experiments in terms of reality and similarity assessment.

| model setup | MovingMnist [33] DFN/SVG/Ours | RobotPush [10] CDNA/SV2P/Ours | Human3.6M [17] DrNet/MCNet/Ours |
|---|---|---|---|
| Reality | 25.4%/43.1%/45.2% | 22.5%/29.2%/35.9% | 10.8%/27.7%/23.1% |
| Image Quality | 28.2%/33.8%/38.0% | 23.5%/30.3%/46.2% | 20.0%/39.2%/40.8% |
| Prediction Accuracy | 38.1%/26.6%/35.3% | 21.9%/36.8%/41.4% | 33.4%/26.0%/40.7% |

hand, this design scheme prevents competition between adversarial and regression losses in principle. Because the gradients back-propagated from them are to optimize different networks, which gets rid of balancing between these two loss functions. On the other hand, sub-networks are designed dedicated for different objectives, *e.g.*, the sampler is required to produce high quality proposals without requirement of motion accuracy, while the selector should be able to fully capture the motion information of previous inputs and select out proposals with high motion accuracy. Note that these objectives are complementary with each other, which essentially encourages corporation between different sub-networks.

**Implementation Details**. We implement the proposed framework with Tensorflow library [1]. The sampler consists of 3 down-sampling Res-Blocks [15] and 3 up-sampling Res-Blocks, where sampling operation is bilinear interpolation with stride 2. Each block is followed by ReLU [12] activation. The kernel generator consists of 5 convolution layers with leaky-ReLU [27] (Leaky rate 0.1) and final layer with sigmoid. The selector is a 3-layer ConvLSTM [32] with stride 4. Finally the combiner consists of two parts, *i.e.*, the encoder and decoder. Encoder part is a 4-layer convolution network with down-sampling of stride 2. The decoder is a mirrored version of the encoder with 4 deconvolution layers and a sigmoid output layer. The model $\varphi$ is identical to that used in style transfer [11]. As mentioned above, we pre-train $\phi_{spl}$ and $\psi_{slt}$ for 2 epochs. The curriculum learning is essentially increasing the prediction length by 1 every epoch with initial length of 1. In all experiments we train all our models with the ADAM optimizer [23] and learning rate $\eta = 0.0001$, $\beta_1 = 0.5$, $\beta_2 = 0.999$. In all experiments we set $N = 5$ and $K = 2$. This selection will be further discussed in Section 4.4.

## 4 Experiments

### 4.1 Datasets and Evaluation Setup

**Datasets**. We evaluate our framework (VPSS) on three diverse datasets: MovingMnist [33], Robot-Push [10] and Human3.6M [17], which represent challenges in different aspects. MovingMnist [33] contains one bouncing digit, which is treated as toy example and suitable for demonstrating the superiority of our framework. RobotPush [10] involves complex robotic motion which has been widely used for video prediction. Human3.6M [17] captures single human motion whose challenge lies in motion stochasticity. Notably, in Human3.6M [17] the human subject only takes a small portion of current frame, whose motion could easily be ignored only with the regression loss.

**Evaluation Setup**. A long-standing issue in video prediction or generation task is how to ensure fair comparison with other models [25]. We try to get rid of unfair comparison by taking following two measures: (1) we compare our model with models whose source codes can be accessed, (2) we exactly follow their experiment setup and do not change their training procedure. For detailed evaluation setup, please refer to Table 2.

### 4.2 Quantitative Evaluation

To quantitatively evaluate our proposed framework, we compute the PSNR and SSIM value for DFN [19], SVG [7] and our model on MovingMnist Datasets [33]. Due to the stochastic prediction of SVG [7], we plot these two curves on average 20 samples for it. Note that different from predicting 10 frames during training, we predict 20 frames to validate the generalization ability. As illustrated

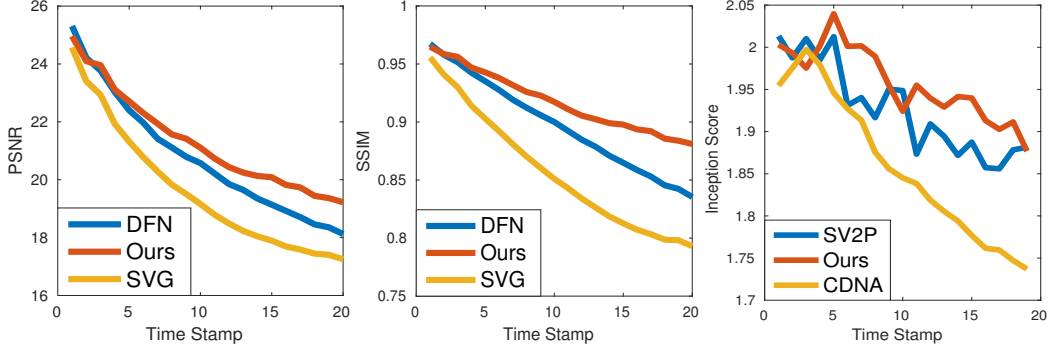

Figure 4: Quantitative comparison with different prediction models in term of PSNR (left), SSIM (middle) and Inception Score (right).

Table 4: Evaluation under PSNR and Inception Score with different combinations of $N$ and $K$

| $(N, K)$ | $(3, 1)$ | $(4, 1)$ | $(3, 2)$ | $(4, 2)$ | $(5, 2)$ |
|---|---|---|---|---|---|
| (PSNR, Inception Score) | (24.43, 1.85) | (24.96, 1.90) | (25.07, 1.95) | (25.21, 1.98) | (25.13, 2.01) |

in Figrue 4.4 (left and middle), our model outperforms the other two by a large margin, especially in latter time stamps. DFN [19] is trained only under regression loss, without modelling the future uncertainty. The prediction results degenerate to blurry frames rapidly. SVG [7] models the future with random noise sampled from a learned prior, which effectively prevents the blur effects. However the correlation between random noise and current inputs is quite weak, which is insufficient to predict the future precisely. In other words, it is more like "random guess". So predictions of SVG [7] are commonly of high image quality but low prediction precision. By contrast within proposed framework both objectives degrade more gracefully than SVG [7] and DFN [19]. Our framework first samples high quality proposals then combines them into final prediction, which effectively tackles above two problems. It clearly proves that the two-stage framework with dedicated designed sub-networks unifies both adversarial and regression loss functions into prediction system successfully.

Recently several works [29, 8, 25] argue that PSNR and SSIM are not convincing enough to guarantee the quality of video prediction. To this end, we compute the Inception Score for CDNA [10], SV2P [3] and our model on RobotPush Datasets [10]. As shown in Figure 4.4 (right), compared to CDNA [10] and SV2P [3] our model keeps relative higher scores throughout the prediction procedure, which mainly benefits from high quality proposals during the sampling stage. The results of CDNA [10] and SV2P [3] clearly demonstrates that balance between loss functions is hard to satisfy all requirements.

### 4.3 Qualitative Evaluation

In video prediction task, the most convincing way to demonstrate effectiveness of model is directly visualizing predicted results. We present several samples predicting up to 20 time stamps for all three datasets shown in Figure 5. As mentioned above, image blur and prediction accuracy are still two key issues we care about. We can clearly observe that DFN [19] and CDNA [10] produce blurry results because of regression loss. While for SVG [7] and SV2P [3], image quality is much better, but compared to ground truth, the prediction accuracy is not so satisfying (*e.g.*, location prediction in MovingMnist [33]). For MCNet [34] and DrNet [8], although they try to enhance the image quality with adversarial learning, the conflict between adversarial loss and regression loss prevents them from achieving both requirements concurrently. By contrast, the proposed two-stage framework achieves both high image quality and precise motion prediction. **We further consider that still images are not sufficient to fully demonstrate information contained in video sequences, so we strongly recommend readers to refer to video results in supplemental material.**

Meanwhile we provide a subjective experiment for further validation. To be specific, we collect 1270 prediction results of each dataset, and ask 40 people to provide subjective assessment on them. This experiment is conducted in three aspects: (1) Regarding the real video samples as baseline, which one is more realistic (Not based on previous inputs)? (2) Considering image quality and previous inputs, which one is more similar to the Ground Truth? (3) Considering motion accuracy and previous inputs, which one is more similar to the Ground Truth? These results are shown in Table 3. In the reality experiment, we observe that results produced by stochastic prediction (SVG [7]) and adversarial learning (Ours and MCNet [34]) related methods seem more realistic to human, which demonstrates

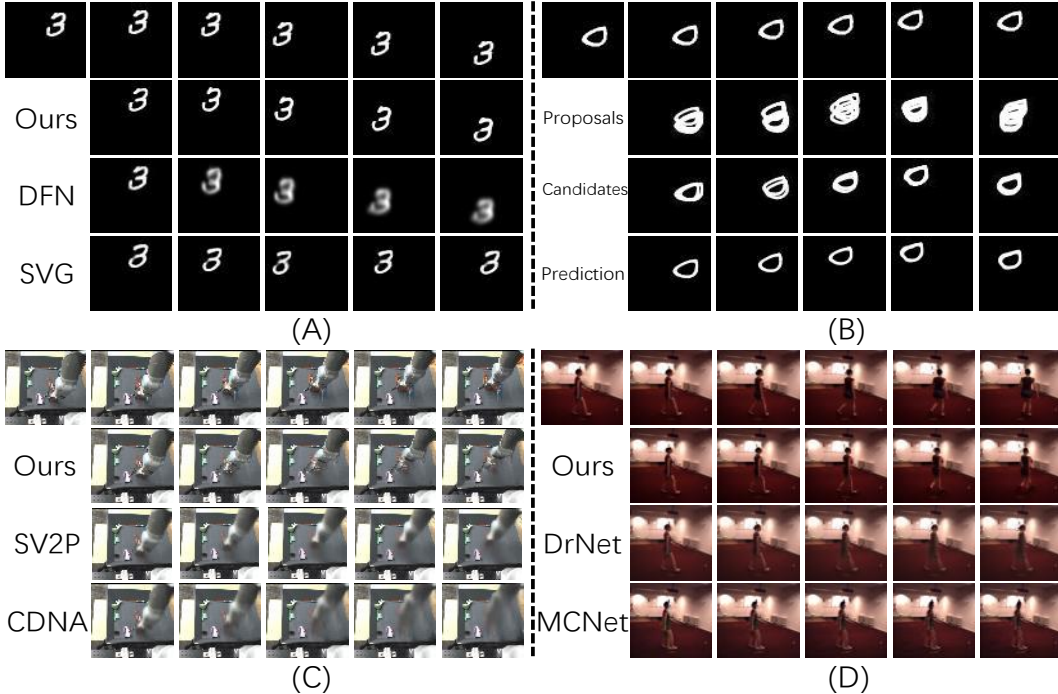

Figure 5: Prediction results on MovingMnist (A) [33], RobotPush (C) [10] and Human3.6M (D) [17] Datasets at time stamp 4,8,12,16,20. Note that sub-figure (B) demonstrates proposals (second row) and candidates (third row) during a complete procedure of predicting moving digit 0. **We strongly recommend readers to refer to more examples in supplemental material.**

effectiveness on enhancing the image quality. Similar effect can be observed even based on previous inputs (Image quality experiment). For the motion accuracy experiment, one can notice considerable preference drop of SVG [7], MCNet [34] and DrNet [8] compared to image quality experiment. Notably, our framework achieves most of the highest scores in three experiments, which mainly benefits from proposed selective sampling framework.

## 4.4 Discussion

Previous experiments demonstrate the superiority of proposed framework. In this section we present further analysis about it. We firstly discuss the selection of $N$ and $K$, then delve deeper into the execution procedure of our framework.

**Selection of $N$ and $K$.** As shown in Table 4, we evaluate the choice of $N$ and $K$ in term of PSNR and Inception Score [10] on Human3.6M Datasets [17]. One can notice that with the increasing of $K$, the prediction accuracy keeps growing (*i.e.*, higher PSNR value). While with the increasing of $N$, the image quality seems to be better (*i.e.*, higher inception score). But keeping $N$ and $K$ too high will drastically increase the model complexity. We choose $N = 5$ and $K = 2$ for the reason that this combination could achieve promising performance and keep model at a relative low complexity level.

**Are these proposals rational enough?** To examine the rationality of proposals at sampling stage, we plot all these proposals in single image for visualization on MovingMnist Datasets [33]. As shown in the second row of Figure 5 (B), the operation of sampler is actually sampling $N$ examples based on previous inputs, and motion direction of all proposals is roughly towards the ground truth. Meanwhile one can observe that the image quality of all proposals keeps at a relative high level and almost does not degrade along the prediction procedure.

**Are these candidates accurate enough?** To examine the accuracy of candidates at selection stage, same as above we plot all these candidates for visualization. As shown in the third row of Figure 5 (B), the operation of selector is actually selecting $K$ candidates from $N$ proposals. For MovingMnist [33] prediction, the selector is actually filtering out these proposals which are possibly far away from the Ground truth. With the help of combiner, high-quality candidates are composed into the final prediction which demonstrates high motion accuracy.

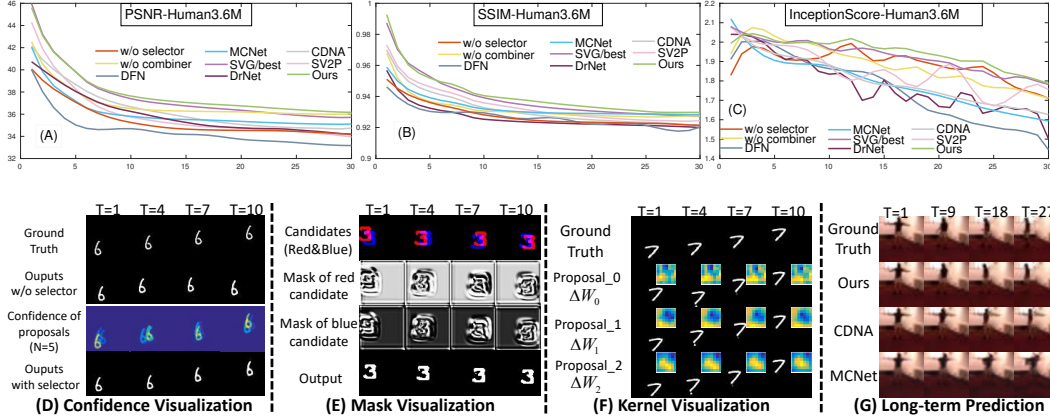

Figure 6: Comparison experiments and ablation study. **w/o: removing corresponding module.**

**Explanation on** $\Delta W$**:** (1) $W$ is the normal learnable kernel and $\Delta W$ is generated at each time stamp. "*Element-wise addition*" means matrix addition between $\Delta W_i$ and $W_i$. (2) We update $\Delta W_i$ instead of $W_i$ for the reason that $W_i$ is treated as a basic kernel which estimates motion at a coarse level, and $\Delta W_i$ is for fine-grained prediction based on $W_i$. By updating $\Delta W_i$ we could narrow down the possible distribution to more precise and smaller scale. (3) Inspired from DFN [9], we use softmax to encourage sparsity of $\Delta W_i$, thus we can mimic the complex motion dynamics more precisely. (4) We present visual evidence (Fig 6(F)) to show the difference among $\{\Delta W_i\}_{i=0}^2$ (2nd input and 2nd output channel of $\{\Delta W_i\}_{i=0}^2$ on up-right corner of each proposal). They also gradually change along with time increasing.

**Comparison Experiments:** As suggested we compare our model with all 6 baselines on Human3.6M datasets [17] in terms of PSNR, SSIM and Inception Score (Fig 6(A,B,C)). Our model (green line) still performs better than these baselines in both terms of motion accuracy and image quality when prediction length is extended to 30 (Long-term prediction, trained for 10 steps). Particularly it is slightly better than the baseline SVG/best (best of 10 random samples, light purple line).

**Ablation Study:** (1) As shown in Fig 6(A,B), when removing selector (K=N) or combiner (K=1), the prediction accuracy drops by a large margin (yellow and orange line) compared to the full model (green line). (2) The predicted results (Fig 6(D)) without selector (2nd row) tend to involve random motion. By contrast, the confidence score (3rd row) assigned by **selector could well estimate the distribution of future motion**, where selector acts as a strong supervisor for motion prediction. (3) The **masks from combiner** (Fig 6(E)) actually combine different parts of candidates for final prediction, which **help to refine the predicted motion** and improve accuracy. (4) Analysis on $N$ and $K$: Further increasing $K$ involves proposals with low confidence, which may potentially decrease the prediction accuracy. Keeping $N$ high will improve performance a little but drastically increase the model complexity.

## 5 Conclusion

In this paper we propose a two-stage framework, called VPSS to study video prediction task from a novel view. At the sampling stage, our framework contains a conditional sampling module which produces multiple high-quality proposals at time each stamp. For the selection stage, we propose a selection module for final prediction. Extensive experiments on diverse challenging datasets demonstrate the effectiveness of the proposed video prediction framework.

## Acknowledgement

This work was supported by National Key Research and Development Program of China (2016YFB1001003), NSFC (61527804, U1611461, 61502301, 61521062). The work was partly supported by State Key Research and Development Program 18DZ2270700. This work was supported by SJTU-UCLA Joint Center for Machine Perception and Inference. The work was also partially supported by China's Thousand Youth Talents Plan, STCSM 17511105401, 18DZ2270700 and MoE Key Lab of Artificial Intelligence, AI Institute, Shanghai Jiao Tong University, China.

## Footnotes

[2]MCNet: https://github.com/rubenvillegas/iclr2017mcnet; DrNet: https://github.com/edenton/drnet; SAVP: https://github.com/alexlee-gk/video_prediction

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
