[Reviews · NeurIPS 2018]

Reviewer 1



The authors propose a method for video prediction with a particular focus on the problem of minimizing blur in predicted frames while optimizing for precision in motion forecasting. The main contribution is an introduced sampling module that explicitly separates different modes in the motion component and thus ensures disentangling the uncertainty in possible futures from visual quality of predicted frames. It is an interesting and to the best of my knowledge original work, both in terms of the theoretical framework and the obtained results. The authors test their implementation on three commonly used benchmarks (MovingMNIST, RobotPush and Human3.6M) by predicting ~10 frames in the future and show significant visual improvements compared to the state-of-the-art, both in terms of visual quality and plausibility of movements. Overall, it was an enjoyable read - however, the clarity of the manuscript can be improved, as at this point some parts are rather difficult to parse. It also contains a certain number of typos and minor grammar mistakes (some of them are listed below). In general, the necessity of each of the proposed blocks could be more explicitly explained and justified by ablation studies. The "combiner" module, for example, should be better motivated: the principled difference between the top candidates and the final result is unclear. What is the meaning of masks in this case and how do they look in practice? It would also be interesting to analyze the performance, possible failure modes and the nature of learned conditional distributions of the selector module - for now there is no discussion on this in the paper. I'm not totally convinced that the experiment described in Table 1 is necessary - these results are rather intuitive and expected. It could make sense to sacrifice it in the interest of space and give more detailed description of the main contributions. The experimental studies are relevant but not exhaustive. It looks like the authors do not provide any quantitative evaluation and comparison with the state-of-the-art on the mentioned Human3.6M dataset. For RobotPush, the only quantitative comparison is expressed in inception scores. Figures 1-3 would be more helpful if the authors added notations used in the text. There is something funny happening with notations for time stamps. L93-94 state the problem as "given x_t, predict x_{t+1}". However, eq. (1) contains x_t and x_{t-1} instead. The next paragraphs (L106-113) talk about all three of them: x_{t-1}, x_t and x_{t+1}. Eq. (5) should probably read k=min|x_{t+1}-\hat{x}_{t+1}| (otherwise, it points to a degenerate solution \hat{x}_t=x_t). Assuming that in reality two frames x_{t-1} and x_t are given to predict the third frame x_t, one wouldn't expect the network to generalize beyond linear motion - is it indeed the case? In practice, does extending the temporal dimension of the input affect the accuracy of motion estimation in longer term? The relationship between W, \Delta W and proposals X is not clear to me. Figure 2 hints that W are convolutional kernels of the sampling decoder (but it's a 3-layer ResNet, according to Section 3.3) but doesn't show how Ws are generated. There is an arrow saying "Element-wise addition" but is has a single input \Delta W_i. If active components of W are different for different proposals \hat{x}_t^i, what prevents us from learning them directly using eq. (5) by selectively updating sub-bands W_i? Symbol W is overloaded and stands for both frame width and kernels. The discriminator in Eq. (1) optimizes for "realism" of proposals x_t regardless of their "compatibility" with the past frames x_{t-1}. I realize this compatibility aspect is taken into account later by predicting "confidence scores", but I'm curious how conditional and unconditional discriminators would compare in this setting (if the authors have experimented with this). L144: why exactly is the per-pixel L1 norm better applicable to linear motion? Section 3.3. The authors mention curriculum learning "along the temporal dimension" but do not explain how it is implemented in their framework. Table 2, third column: it's not clear what is meant by "inputs" and "outputs" (are the inputs processed / outputs predicted at once or in a sliding window manner? is it the same for all frameworks? how is it compatible with the description in Section 3?) How do the authors explain that the performance on Human3.6M simultaneously goes down in terms of "reality" (23.1%) but improves in terms of "image quality" (40.8%)? (Table3, last column). Section 4, Figure 4. Could the authors comment on why the proposed method is compared to DFN and SVG on Moving MNIST (in terms of PSNR and SSIM), but to SV2P and CDNA on RobotPush (in terms of inception scores)? It would be also interesting how performance of these methods compares to trivial baselines (copying the last observed frame, warping with optical flow). L262-267. Which dataset do the author refer to? Please also mention it in the caption of Table 4. The performance is improved by increasing K from 1 to 2 - what happens when N and/or K grows further? Again, I would like to see more empirical evidence justifying the presence of each component: can we remove the selector (i.e. set K=N)? or the combiner? Nitpicking: L30: is is possible -> it is possible L36: As show -> As shown L48: Nitish et al. -> Srivastava et al. L49: [32] propose -> [32] proposes L54: methods suffers -> method suffer L61: inference is not a verb L76: 34 twice L117: Rivera et al. -> Guzman-Rivera et al. L228: As show -> as shown Ref [23]: CoRR -> ICLR, 2014 Ref [29]: CoRR -> ICLR, 2016 etc.

Reviewer 2



Summary: This paper proposes a method for future frame prediction that consists of a sampling stage in which possible future frames given the past are generated by a set of weight matrices that model future dynamics based on the difference of the previous two consecutive frames. The best candidate future frames are then chosen based on a confidence score computed by an RNN. The sampling stage is then followed by a combination layer that fuses the best candidates into a single future frame. Their objective function consists for a combination of a selection based adversarial training on the generated image that is closest to the ground truth and regression loss on the RNN output scores between the generated future frames and the ground truth frames. In experiments, they evaluate and compare their method in terms of pixel based and human judgement evaluations. They also provide video comparison against the baselines showing the advantage of the proposed method. Pros: Novel architecture that makes sense Numerically and visually outperforms baselines Paper is overall easy to understand Cons: Intuition/explanation: Why is \Delta W output instead of simple W for the kernels? Why is a softmax activation used for \Detal W? How do the authors make sure that the \Delta Ws are different? The argument given in the paper is that they are updated in different directions, but there is no concrete evidence that this is actually happening. It would be good if the authors give a better intuition or show evidence of their claims in these cases. Comparison against SVG: The authors mention that they plot the curve that is an average of 20 samples from that method. Having 20 samples is too little for the average to be significant. How about the authors compare against the best of the 20 samples? Or reduce the number of samples and still compare against the best. If the proposed method is still better against such optimistic baseline, it would be a very nice result, and would high-light the advantage of the proposed architecture. Ablative studies: If possible, it would be good if the authors can show an ablative study of how each component in the proposed network helps (e.g., instead of combination masks, just average the best samples, no best k samples selection, etc) Discussion: The authors show some visual analysis of the proposals based on being “rational enough”, “accurate enough”. This analysis is shown on a single example. I believe this analysis should be done in a different (hopefully numerical way), as one example is not very informative. Typos: The paper needs to be proofread. There are a few typos. Overall, I like the paper, however, there are some issues as mentioned above that the authors should take into consideration.

Reviewer 3



== Summary == The paper tackles the video generation problem by a selective sampling mechanism. Image blur and motion deformation are two big challenges in video prediction. Previous video generation work optimizes a hybrid objective, which struggles to balance between adversarial and regression objective. To solve the issue, this paper consider a two-stage framework the decouples the adversarial and regression loss into separate modules. In the sampling stage, a kernel generation network produces multiple convolutional kernels, while a sampler network applies kernels to predict the next frames. In the selection stage, a selector networks finds the top K predictions under the measure of motion precision, while a combiner network composes the final prediction from K candidates. Experimental evaluations are conducted on three benchmark datasets: MovingMNIST, RobotPush, and Human3.6M. == Quality and Originality == Overall, the proposed VPSS method is a promising solution to the video prediction problem. The proposed 2-stage pipeline improves the prediction quality (e.g., low-level measurements and perceptual realism) to some extent. One drawback is that the proposed 2-stage pipeline is a bit complicated. Furthermore, reviewers do have some reservations on the experimental design. Please address the issues mentioned below in the rebuttal. == Generating long-term future == As improved image quality is the main contribution of the VPSS method, it would be more convincing to demonstrate the long-term prediction results and compare with existing methods. Reviewer would like to see, for example, the 20 steps prediction results on Human3.6M dataset using the model trained for 10 steps. == Side-by-side comparisons == Reviewer would like to see the side-by-side comparisons to all baseline methods (DFN/SVG/CDNA/SV2P/DrNet/MCNet) on at least one dataset. The current experimental results look less solid since only two baselines are compared on each dataset. For example, DFN is better in prediction accuracy than VPSS on MovingMNIST dataset. == Implementation Details == As the 2-stage pipeline is a bit complicated, it becomes non-trivial to reproduce the results. Also, reviewer would like to know whether the proposed VPSS is efficient or not in terms of training time and inference time compared to video prediction baselines. **** Regarding Author Feedback **** The rebuttal resolves the following concerns: -- Side-by-side comparisons -- Long-term prediction Reviewer 3 agrees with other reviewers that this submission is a good candidate for acceptance. It would be great if the ablation study is included in the final version. Reviewer 3 does not feel strong motivations to raise the score, as the proposed 2-stage method is a bit complicated. Reviewer encourages the authors to explore a simplified version along this direction as future work.